# Anterior cingulate cortex causally supports flexible learning under motivationally challenging and cognitively demanding conditions

Kianoush Banaie Boroujeni[1]*, Michelle K. Sigona[2,3], Robert Louie Treuting[3], Thomas J. Manuel[2,3], Charles F. Caskey[2,3,4], Thilo Womelsdorf[1,3]*

1 Department of Psychology, Vanderbilt University, Nashville, Tennessee, United States of America, 2 Vanderbilt University Institute of Imaging Science, Nashville, Tennessee, United States of America, 3 Department of Biomedical Engineering, Vanderbilt University, Nashville, Tennessee, United States of America, 4 Vanderbilt University Medical Center Department of Radiology and Radiological Sciences, Nashville, Tennessee, United States of America

* kianoush.banaie.boroujeni@vanderbilt.edu (KBB); thilo.womelsdorf@vanderbilt.edu (TW)

**Data Availability Statement:** All data supporting this study and its findings can be find using the link: https://figshare.com/projects/TUS_

## Abstract

Anterior cingulate cortex (ACC) and striatum (STR) contain neurons encoding not only the expected values of actions, but also the value of stimulus features irrespective of actions. Values about stimulus features in ACC or STR might contribute to adaptive behavior by guiding fixational information sampling and biasing choices toward relevant objects, but they might also have indirect motivational functions by enabling subjects to estimate the value of putting effort into choosing objects. Here, we tested these possibilities by modulating neuronal activity in ACC and STR of nonhuman primates using transcranial ultrasound stimulation while subjects learned the relevance of objects in situations with varying motivational and cognitive demands. Motivational demand was indexed by varying gains and losses during learning, while cognitive demand was varied by increasing the uncertainty about which object features could be relevant during learning. We found that ultrasound stimulation of the ACC, but not the STR, reduced learning efficiency and prolonged information sampling when the task required averting losses and motivational demands were high. Reduced learning efficiency was particularly evident at higher cognitive demands and when subjects experienced loss of already attained tokens. These results suggest that the ACC supports flexible learning of feature values when loss experiences impose a motivational challenge and when uncertainty about the relevance of objects is high. Taken together, these findings provide causal evidence that the ACC facilitates resource allocation and improves visual information sampling during adaptive behavior.

## Introduction

It is well established that the anterior cingulate cortex (ACC) and the anterior striatum contribute to flexible learning [1–4]. Widespread lesions of either structure can lead to

PlosBiology/144330. Custom MATLAB code generated for analyses, are available from the GitHub link: https://github.com/banaiek/TUS_PlosBiology_2022.git.

**Funding:** This work was supported by the National Institute of Mental Health of the National Institutes of Health under Award Number R01MH123687 (TW) and 1UF1NS107666 (CC). The funders had no role in study design, data collection and analysis, decision to publish, or preparation of the manuscript.

**Competing interests:** The authors have declared that no competing interests exist.

**Abbreviations:** ACC, anterior cingulate cortex; GLM, generalized linear model; GTI, gross token income; LME, linear mixed effect; MI, modulation index; STR, striatum; tDCS, transcranial direct current stimulation; TMS, transcranial magnetic stimulation; TUS, transcranial ultrasound stimulation.

nonadaptive behavior. When tasks require subjects to adjust choice strategies, lesions in the ACC cause subjects to shift away from a rewarding strategy even after having obtained reward for a choice [5] and reduce the ability to use error feedback to improve behavior [6–8]. Lesions in the striatum likewise reduce the ability to use negative error feedback to adjust choice strategies, which leads to perseveration of non-rewarded choices [9], or inconsistent switching to alternative options after errors [10]. A common interpretation of these lesion effects is that both structures are necessary for integrating the outcome of recent choices to update the expected values of possible choice options. According to this view, the ACC and striatum keep track of reward outcomes of available choice options in a given task environment.

However, it has remained elusive what type of reward outcome information is tracked in these structures and whether outcome information in ACC or the striatum affects learning even when it is not associated with specific actions. In many learning tasks used to study ACC or striatum functions, subjects learned associating reward outcomes with the direction of a saccadic eye movement or the direction of a manual movement [11–14]. Succeeding with these tasks requires computing probabilities of action-reward associations. However, recent studies have documented neurons in ACC and striatum that not only tracked action-reward association probabilities, but also the expected reward value of specific features of chosen objects [11,15–19]. Neuronal information about the value object features emerged slightly earlier in ACC than in striatum [15,16,19] and rapidly synchronized between both structures suggesting they are available across the frontostriatal network at similar times [20].

These findings suggest that the ACC or striatum may be functionally important to learn expected values of objects' specific visual features and thereby mediate information-seeking behavior and visual attention [17,21–25]. There are at least 3 possible ways how feature-specific value information in these areas may support flexible learning. A first possibility is that either of these brain areas uses feature-specific value information for credit-assigning reward outcomes to goal-relevant object features. Such a credit assignment process is necessary during learning to reduce uncertainty about which features are most relevant in a given environment. Support for this suggestion comes from studies reporting of neurons in ACC that show stronger encoding of task variables in situations with higher uncertainty [26], respond to cues reducing uncertainties about outcomes [27], and form subpopulations encoding uncertain outcomes [16]. These prior studies predict that ACC or striatum will be important for learning values of object features when there is high uncertainty about the reward value of object features.

A second possibility is that the ACC or striatum uses feature-specific value information to determine whether subjects should continue with a current choice strategy or switch to a new strategy [28,29]. According to this framework, the ACC's major role is to compute, track, and compare an ongoing choice value for going on with similar choices or switching to alternative choice options. This view predicts that when errors accumulate during learning, the estimated choice value for continuing similar choices is reduced relative to alternatives and ACC or striatum will activate to either directly modify choice behavior [28,29] or indirectly affect choices by guiding attention and information sampling away from recently chosen objects and toward other, potentially more rewarding objects [25].

A third possible route for value information in ACC and striatum to affect behavior assumes that information about the value of object features is not used to compute a choice value, but a motivational value for the subject that indicates whether it is worth to put continued effort into finding the most valuable object in a given environment [4,30]. According to such an effort-control framework, value signals across the ACC-striatum axis are used to compute the value of controlling task performance [31]. Similar to the choice-value framework, this motivational value framework predicts that ACC and striatum should become more

important for learning when the number of conflicting features increases. While in the choice-value framework, an increase in conflicting features will evoke more ACC activity by increasing the number of comparisons between an ongoing choice value and the value of the other available options, in the effort-control framework, more feature conflict with the same motivational payoff requires more ACC activity to decide whether it is worth to put effort in the task or not [29,30]. Key motivational factors determining whether subjects put more effort into learning even difficult problems are the amount of reward that can be gained or the amount of punishment or loss that can be avoided by putting effort in the task. Support for suggesting an important role of the ACC to mediate how incentives and disincentives affect learning comes from studies showing ACC neurons responding vigorously to negative outcomes such as errors [32], respond to negatively valenced stimuli or events irrespective of error outcomes [33–35], and that fire stronger when the subject anticipates aversive events [27]. These insights suggest that the ACC and possibly the striatum will be relevant for learning particularly when it involves averting punishment or loss.

Here, we set up an experiment to test the relative roles of cognitive and motivational contributions of the ACC and striatum for the flexible learning of feature values. We adopted a transcranial ultrasound stimulation (TUS) protocol to temporarily modulate neuronal activity in the ACC or the striatum of rhesus monkeys while they learned values of object features through trial-and-error. The task independently varied *cognitive load* by varying the number of unrewarded features of objects and *motivational context* by varying the amount of gains or losses subjects could receive for successful and erroneous task performance (**Fig 1A and 1B**). Subjects learned a feature-reward rule by choosing 1 of 3 objects that varied in features of only 1 dimension (low cognitive load, e.g., different shapes), or in features of 2 or 3 dimensions (high cognitive load, e.g., varying shapes, surface patterns, and arm types) (**Fig 1D**). Independent of cognitive load, we varied how motivationally challenging task performance was by altering whether the learning context was a pure *gain-only* context or a mixed *gain-loss* context. In the *gain-only contexts*, subjects received 3 tokens for correct choices, while in the *gain-loss contexts*, subjects received 2 tokens for correct choices and lost 1 already attained token when choosing objects with non-rewarded features (**Fig 1A–1C**). Such a loss experience has been reported in previous studies to impose a motivational conflict [36], inferred also from vigilance responses triggered by experiencing losses [37]. The task required monkeys to collect 5 visual tokens before they were cashed out for fluid rewards.

With this design, we found that sonication of the ACC, but not the anterior striatum, with TUS led to a learning deficit when subjects experienced losses. This loss-triggered deficit was accompanied by inefficient information sampling and was most pronounced at high cognitive load.

## Results

We applied transcranial focused ultrasound (TUS) to modulate neural activity in ACC (area 24) or anterior striatum (STR, head of the caudate nucleus) in 2 monkeys in separate learning sessions by adopting the same TUS protocol as in [38,39]. The sonication protocol imposed an approximately 6-mm wide/40-mm tall sonication region that has been shown previously to alter behavior in foraging tasks [40] to reduce functional connectivity of the sonicated area in macaques [38] and in in vitro preparations to modulate neuronal excitability to external inputs [41]. We provide detailed acoustic simulations of the ultrasound pressure dispersion around the target brain areas, the anatomical sub-millimeter targeting precision of TUS, and the validation of the applied ultrasound power through real-time power monitoring during the experiment in **S3 Fig**. We bilaterally sonicated or sham-sonicated the ACC or the STR in individual

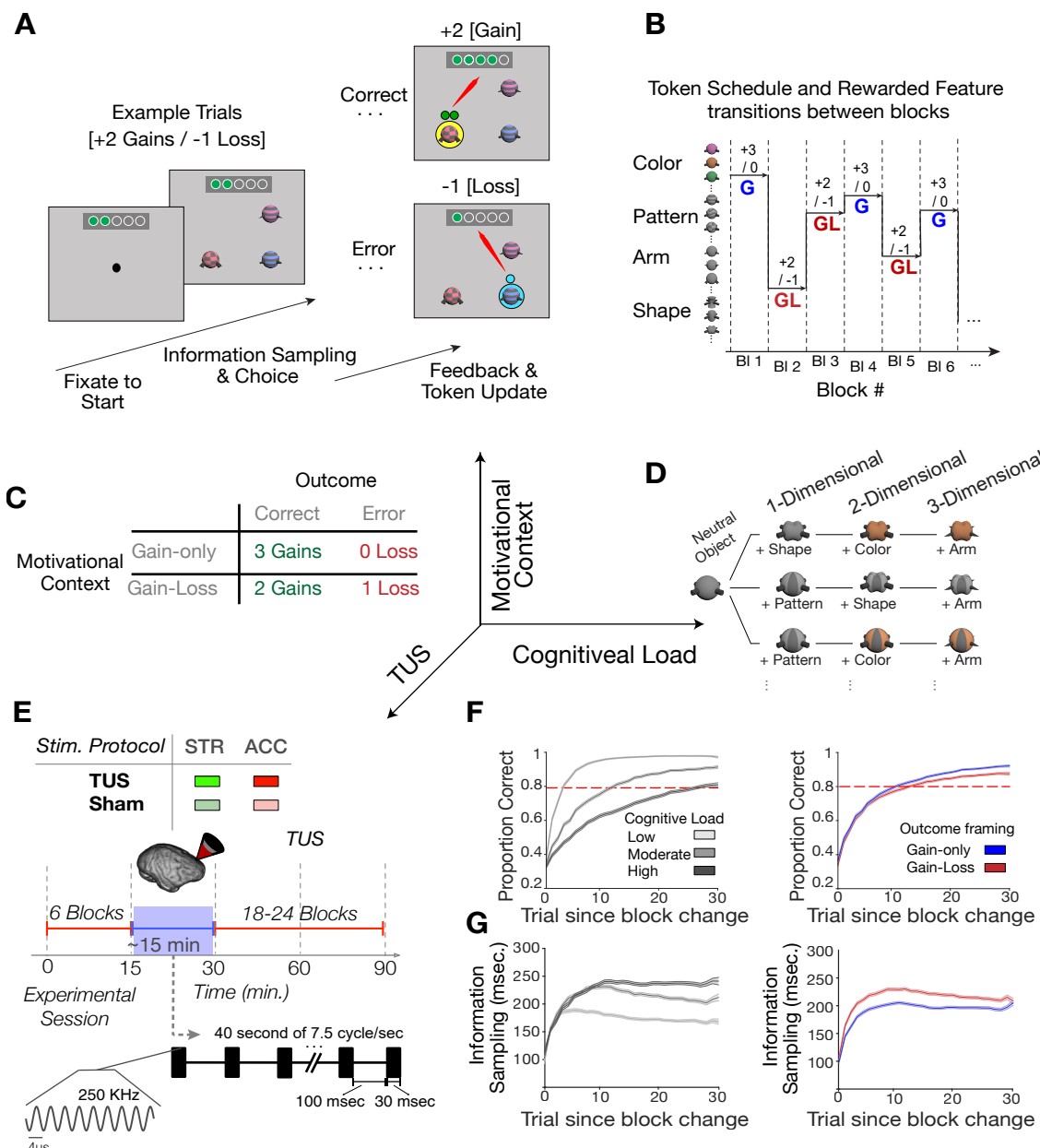

**Fig 1. Task paradigm and TUS protocol.** (**A**) A trial started with central gaze fixation and appearance of 3 objects. Monkeys could then explore objects and choose 1 object by fixating it for 700 ms. A correct choice triggered visual feedback (a yellow halo of the chosen object) and the appearance of green circles (tokens for reward) above the chosen object. The tokens were then animated and traveled to a token bar on the top of the screen. Incorrect choices triggered a blue halo, and in the gain-loss condition 1 blue token was shown that traveled to the token bar where one already attained token was removed. When ≥5 tokens were collected in the token bar, fluid reward was delivered, and the token bar reset to zero. (**B**) In successive learning blocks, different visual features were associated with reward and blocks alternated randomly between the gain-only (G) and gain-loss (GL) conditions. (**C**) In the gain-only motivational context, monkeys gained 3 tokens for correct and 0 penalty for each incorrect choice, whereas in the gain-loss context, they gained 2 tokens for each correct and lost 1 token for each incorrect response. The axes (right) show the 3 orthogonal independent variables of the task design (cognitive load, motivational context, and TUS conditions). (**D**) Cognitive load varied by increasing the number of object features from 1 to 3 and from block to block. (**E**) In each sonication or sham session, the experiment was paused after 6 learning blocks. There are 4 experimental conditions; TUS in ACC (ACC—TUS; red), or anterior striatum (STR-TUS; green), or sham ACC (ACC-Sham; dimmed red), or sham anterior striatum (STR-Sham; dimmed green); 30-ms bursts of TUS were delivered every 100 ms over a duration of 80 seconds (40 seconds each hemisphere). (**F**) Proportion of correct choices over trials since block begin for different cognitive loads (left panel; 1–3D, light to dark gray) and motivational contexts (gain-only, blue; gain-loss, red). (**G**) The average fixation duration on objects prior to choosing an object (*information sampling*) in the same format as in (**F**). The lines show the mean and the shaded error bars are SE. Data associated with this plot could be found at: https://figshare.com/projects/TUS_PlosBiology/144330. ACC, anterior cingulate cortex; STR, striatum; TUS, transcranial ultrasound stimulation.

sessions immediately after monkeys had completed the first 6 learning blocks. We performed 12 experimental sessions for each TUS or sham condition in each of the 2 brain areas ACC or STR (a total of 48 sessions) with each of the 2 monkeys. Following the sonication procedure, monkeys resumed the task and proceeded on average for 23.6 (±4 SE) learning blocks (monkey W: 20.5 ± 4; monkey I: 26.5 ± 4) (**Fig 1E**).

Across learning blocks, monkeys reached the learning criterion ($\geq$80% correct choices over 12 trials) systematically later in blocks with high cognitive load (linear mixed effect (LME) model with a main effect of cognitive load, $p < 0.001$; **Figs 1F and** S1A–S1C). Both monkeys also showed longer foveation durations onto the objects prior to making a choice when the cognitive load was high (LME main effect of cognitive load, $p < 0.001$; **Figs 1G and** S1D **and** S1E). Longer foveation durations index more extensive information sampling of object features at higher cognitive load. We defined information sampling as the duration monkeys fixated objects prior to the last fixation in a trial that was used by the subjects to choose an object. This metric indexes how long information was processed about feature values of the objects prior to committing to a choice. Monkeys also increased information sampling, significantly slowed learning, and showed reduced plateau performance (**S2A–S2C Fig**) in blocks with gains and losses (gain-loss contexts) compared to blocks with only gains (gain-only contexts) (**Figs 1G and S2D and S2E**; LME main effect of motivational context, $p < 0.001$).

TUS in ACC (ACC-TUS) but not sham-TUS in ACC (ACC-Sham) or TUS in striatum (STR-TUS) or sham-TUS in the striatum (STR-Sham) (**Fig 2A**) changed this behavioral pattern. TUS conditions showed a significant interaction with motivational context with ACC-TUS selectively slowing learning in the gain-loss contexts compared to the gain-only contexts (LME interaction of TUS condition and motivational context, t = 2.67, $p = 0.007$) (**Figs 2B and** S4A **and** S4B **and S1 Table**). ACC-TUS increased the number of trials needed to reach the learning criterion of 80% performance to 14.7 ± 0.8 trials (monkey W/I: 14.3 ± 1.2/ 15 ± 1.2) relative to the pre-TUS baseline (trials to criterion: 10.7 ± 1.4; monkey W/I: 9.9 ± 2.3/ 11.4 ± 1.8) (Wilcoxon test, $p = 0.049$; **Figs 2B and** S4C and S2D). The learning speed with ACC-TUS in the gain-loss context was significantly slower than in other TUS conditions (Kruskal–Wallis test, $p = 0.003$), ACC-Sham (pairwise Wilcoxon test, FDR multiple comparison corrected for dependent samples, $p = 0.019$), and to the conditions STR-TUS and STR-Sham (pairwise Wilcoxon test, FDR multiple comparison corrected for dependent samples, $p = 0.019$, $p = 0.003$) (**Fig 2B**). This effect interacted with cognitive load. The slower learning after ACC-TUS in the gain-loss condition was stronger when cognitive load was intermediate or high, i.e., in conditions with 2 or 3 distracting feature dimensions (random permutation, $p < 0.05$; LME 3-way interaction TUS condition, cognitive load, and motivational context, t = −2.8, $p = 0.004$) (**Figs 3C and 3D and** S5A **and S2 Table**). TUS did not affect learning in the gain-only contexts even at high cognitive load (Kruskal–Wallis test, $p = 0.933$) (**Figs 3C and 3D and** S5B).

TUS conditions showed a significant interaction with motivational conditions on information sampling (LME, t = −4.03, $p < 0.001$). The slower learning in the gain-loss context after ACC-TUS was accompanied by prolonged information sampling compared to the pre-TUS baseline (ACC-TUS information sampling: 234 ± 6 ms monkey W/I: 230 ± 6/237 ± 7 ms, pre-TUS information sampling: 209 ± 11 ms monkey W/I: 197 ± 16/221 ± 11 ms; Wilcoxon test, $p = 0.016$) and compared to STR-TUS (Kruskal–Wallis test, $p = 0.036$; pairwise Wilcoxon test, FDR multiple comparison corrected for dependent samples, $p = 0.03$) (**Figs 3A and** S6A **and** S6C). Fixational information sampling in the gain-only context did not vary between TUS conditions (Kruskal–Wallis test, $p = 0.55$) (**Figs 3B and** S6B **and** S6D; for detailed information about the distribution of fixational information sampling and its bootstrap sampling distributions for different motivational contexts, cognitive load, and TUS conditions, see **S7 Fig**).

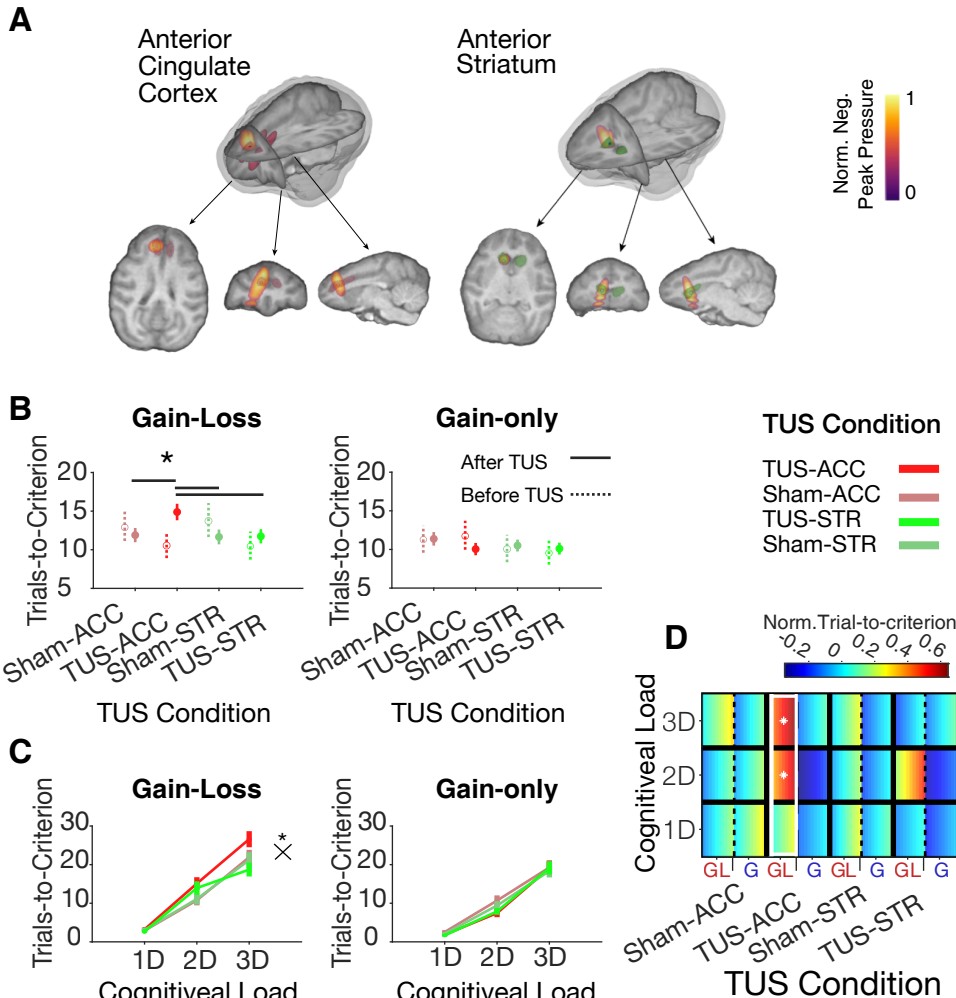

**Fig 2. TUS of ACC and anterior striatum.** (**A**) The maximum negative peak pressure of TUS in an example session of 1 monkey shows the focus is within ACC (left) and anterior striatum (right). (**B**) Learning is slowed down (trials-to-criterion increased) after TUS in ACC in the gain-loss context (left, LMEs, $p = 0.007$) but not in the gain-only contexts (right, n.s.). Data represent means and the standard error of the mean. (**C**) Learning is slowed with higher cognitive load with a significant interaction in the gain-loss context (left, LMEs, $p = 0.007$) but not with the gain-only context (right, n.s., for the full multiple comparison corrected statistical results, see **S1 and S2 Tables**). (**D**) Marginally normalized trials-to-criterion is significantly higher with ACC-TUS in the gain-loss (GL) learning context at higher (2D and 3D) cognitive load (random permutation $p < 0.05$). Each cell is color coded with the mean value ± SE with a low to high value gradient from left to right. Values across learning blocks in each cell are normalized by subtracting the mean and dividing by standard deviation of all baseline learning blocks across all TUS conditions for each load and motivational context. The white rectangle indicates that learning in that TUS condition (*x-axis*) is different from other TUS conditions. White asterisks indicate that a cognitive load condition in a learning context in a TUS condition is significantly different from other TUS conditions. Black crosses (×) and asterisks mark significant interactions of cognitive load and TUS conditions. Black asterisks indicate significant main effects of TUS conditions and a significant difference between the TUS and the baseline (pre-TUS) conditions (for the TUS condition underneath the asterisks). Horizontal black lines indicate significant pairwise differences between TUS conditions. Data associated with this plot could be found at: https://figshare.com/projects/TUS_PlosBiology/144330. ACC, anterior cingulate cortex; LME, linear mixed effect; TUS, transcranial ultrasound stimulation.

Once subjects reached the learning criterion in a learning context, they could exploit the learned feature rule until the block changed to a new feature rule after approximately 30 to 55 trials. During this period, they showed overall high plateau performance, which was significantly lower in the gain-loss context (87% ± 0.07) than the gain-only contexts (90% ± 0.05,

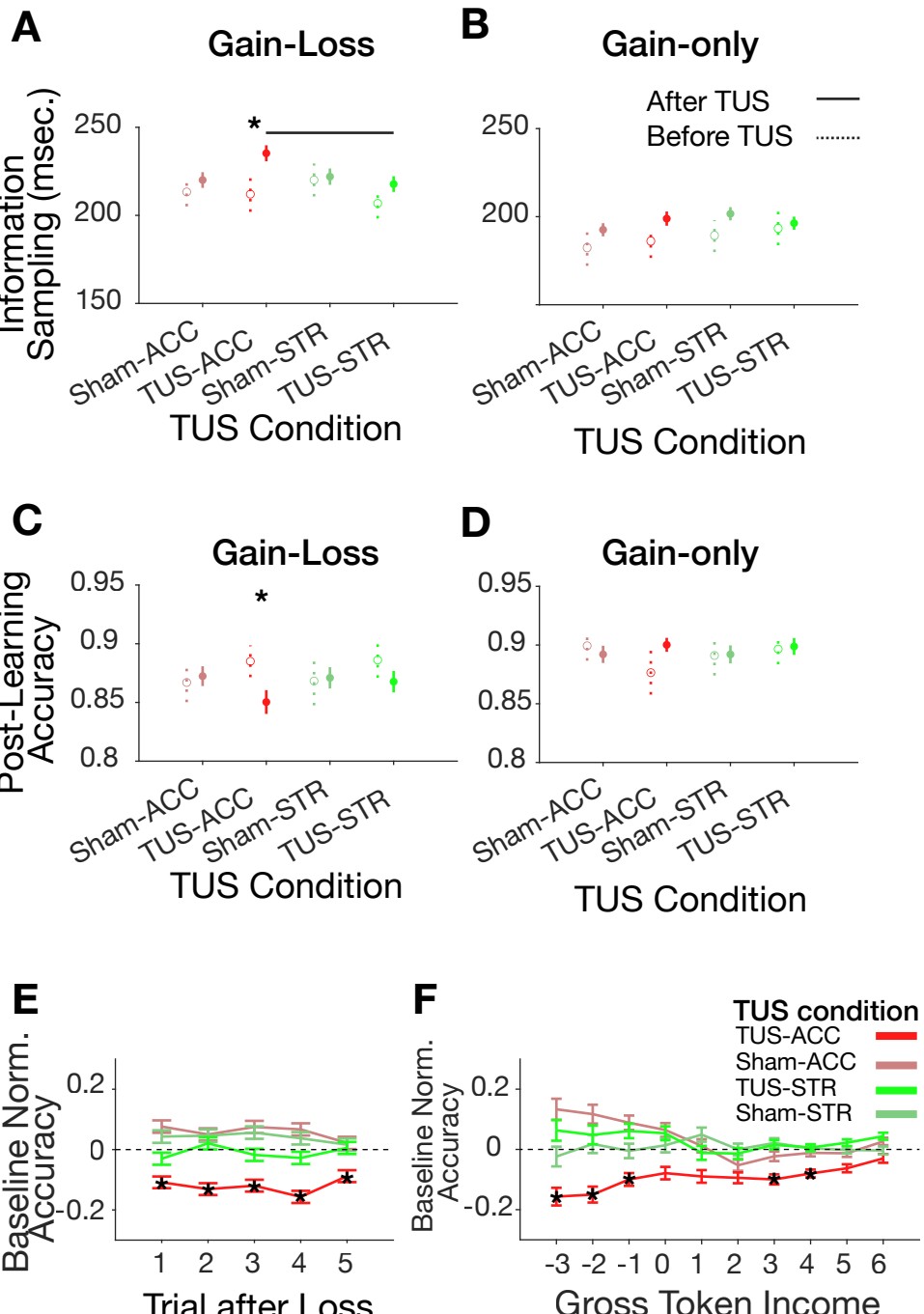

**Fig 3. TUS effect on fixational information sampling and behavioral adjustment after gains and losses.** (**A,B**) Information sampling (the duration of fixating objects prior to choosing an object) is increased with TUS in ACC in gain-loss (**A**) but not gain-only (**B**) learning context. (**C, D**) TUS in ACC reduced post-learning plateau accuracy in gain-loss contexts (**C**) but not in gain-only contexts (**D**) (LMEs, $p = 0.04$) (see **S2 Table**). With FDR correction, this accuracy effect was significant at the session level, but not at the block level, indicating a low effect size (**S10E Fig** and **S1 Table**). (**E**) The accuracy was overall reduced with TUS in ACC in the 5 trials after experiencing a token loss in the gain-loss context (random permutation, $p < 0.05$). (F) GTI (*x-axis*) measures the signed average of tokens gained and lost in the preceding trials. TUS in ACC reduced the performance accuracy (*y-axis*) when monkeys had lost more tokens in the near past (negative GTI) (random permutation, $p < 0.05$). Data indicate the mean and the standard error of the mean. Accuracy on the trial level is normalized by the mean and standard deviation of the accuracy in the baseline (the first 6 blocks prior to the TUS) in the same TUS sessions. Black asterisks show significant main effect of

TUS conditions and a significant difference between the TUS and the baseline (pre-TUS) conditions (for the TUS condition underneath the asterisks). Horizontal black lines indicate significant pairwise difference between TUS conditions. Data associated with this plot could be found at: https://figshare.com/projects/TUS_PlosBiology/144330. ACC, anterior cingulate cortex; GTI, gross token income; LME, linear mixed effect; TUS, transcranial ultrasound stimulation.

LME main effect of motivational context, t = −3.95, $p$ = 0.001) (**S2C Fig**). ACC-TUS exacerbated this performance drop in the gain-loss block, leading to significantly lower plateau accuracy compared to the pre-TUS baseline condition and compared to other TUS conditions in the gain-loss learning context (LME interaction of TUS condition, motivational context and pre/post sonication, t = −2.05, $p$ = 0.04; **Figs 3C and** S6E), but not in the gain-only learning context (Kruskal–Wallis test, $p$ = 0.8) (**Figs 3D and** S6F). The mean plateau accuracy was reduced to 84.5% ± 1.4 (monkey W/I: 85% ± 1.4/84% ± 1.4) relative to the pre-TUS baseline (plateau accuracy: 88 ± 1.8; monkey W/I: 88 ± 2/88 ± 1.6). We confirmed these results using a second metric to estimate learning and plateau performance by fitting logistic general linear models (GLMs) to performance accuracy in each block (all previously significant results for trial-to-criterion and plateau accuracy remained significant when instead comparing inflection point and asymptote, **S8 Fig**).

We validated that the behavioral impairments emerged shortly after the sonication and lasted until the end of the ≤120-min long session (**S9 Fig**). We also confirmed that the observed behavioral effects were not only evident when considering individual blocks (the block-level analysis) (**S1 and S2 Figs**), but also when averaging the learning performance across blocks per session and applying session-level statistics (S10 Fig **and S1 Table**).

So far, we found that ACC-TUS slowed learning speed, increased information sampling durations, and reduced plateau accuracy in the gain-loss contexts. These results could be due to motivational difficulties when adjusting to the experience of losing an already attained token. To analyze this adjustment to losses, we analyzed the performance in trials following choices that led to losses. We found that experiencing a loss leads to overall poorer performance in subsequent trials after ACC-TUS, but not after ACC-Sham, STR-TUS, or STR-Sham (random permutation test, $p$ < 0.05, **Figs 3E and** S11A–S11D). Importantly, this overall performance decrement was dependent on the recent history of losses. ACC-TUS reduced performance accuracy on trials in the gain-loss context, specifically when subjects had lost 2 or 3 tokens in the preceding 4 trials, but not when their net token gain in the past 4 trials was ≥0 tokens, or for trials in gain-only context (random permutation test, $p$ < 0.05, **Figs 3F and** S11). This dependence of the ACC-TUS effect on the recent gross token income (GTI) was evident in both monkeys (**S11E Fig**) and could be a main reason that led to slower learning.

## Discussion

We found that sonicating the ACC, but not the anterior striatum (head of the caudate), slowed down learning, prolonged visual information sampling of objects prior to choosing an object, and reduced overall performance when learning took place in a context with gains and losses and with high cognitive load. These behavioral impairments were specific to the ACC-TUS condition when comparing behavior to baseline performance prior to TUS and to sessions with sham-controlled TUS and STR-TUS. Moreover, the changes in behavioral adjustment were found when comparing average performance between individual sessions (**S10 Fig**), for performance variations across blocks of different sessions (**Fig 2**), and on a trial-by-trial level in impaired behavior when the loss of tokens accumulated over trials (**Fig 3**).

Taken together, these findings provide evidence that primate ACC supports the guidance of attention and information sampling in contexts that are motivationally challenging and cognitively demanding. Prior to discussing the implications of these findings, it should be noted that the sonication effects on neuronal activity in vivo are not well investigated. The sonication protocol we applied may involve changes in the excitability of neural tissue within the hot spot of the sonicated area [41] or effects on the fibers of passage through the sonication areas. Our discussion assumes a putative disruptive effect of TUS on neural activity within ACC. This assumption is based on a prior study that used the same TUS protocol and reported reduced functional connectivity of the sonicated brain area [38]. There is also the caveat that the sonication effects could indirectly follow altered activity in other areas through modulating connectivity with those areas.

## Extending existing functional accounts of ACC functions

Our main findings suggest extensions to theoretical accounts of the overarching function of the ACC for adaptive behavior. First, the finding of prolonged information sampling after ACC-TUS supports recent electrophysiological and imaging studies showing that activity in the ACC predicts information sampling of visual objects and attention to maximize reward outcomes [17,18,25,42]. However, we found a functional deficit of information sampling only in the gain-loss learning context in which subjects expected lower payoff, suggesting that the ACC contributes to information sampling, particularly in motivationally challenging conditions.

Secondly, our core finding of compromised learning in motivationally challenging conditions partly supports and extends the view that ACC is essential to control effort [30]. The functional impairment in the gain-loss motivational context was most pronounced when subjects faced high cognitive load, i.e., in the condition that overall was most difficult. While this result pattern suggests that the overall difficulty may be the primary driver of the ACC-TUS effects, we outline below that the specific impairments after loss experiences (irrespective of cognitive load) and the overall reduced plateau performance with ACC-TUS in the gain-loss context indicate a predominant role of motivational processes over cognitive processes to drive the behavioral ACC-TUS effects. At a psychological level, the pronounced deficit in the loss context at high cognitive load is consistent with a neuroeconomic view of ACC function based on prospect theory, which suggest that losses induce a particularly strong internal demand for adjusting actions, which is intensified when the adjustment of the action itself gets more demanding by an increase in cognitive load [37,43,44]. A particular role of the ACC in mediating the motivational consequences of negative loss experiences is consistent with a wealth of studies about affective processing, mostly from human imaging studies, that show systematic ACC activation in the face of aversive or threatening experiences [33,34,45].

Our finding that experiencing losses was necessary to observe behavioral deficits from ACC-TUS shows that the presence of a cognitive conflict due to an increased number of features in the high cognitive load condition was not sufficient to alter performance. This result might seem at odds with literature implicating the ACC to be particularly relevant to resolving conflict [46,47]. However, increasing cognitive load in our task entailed increased perceptual interference from object features, it did not entail a conflict of sensory-motor mappings that is the hallmark of conflict-inducing flanker, Stroop, or Simon tasks, used to document a role of the ACC to mediate the resolution of conflict [26,46,48]. Therefore, our results do not oppose studies using those paradigms, which gave rise to the view that neuronal signaling in ACC contributes to resolving sensorimotor conflicts [2]. Our results instead point to the importance of the ACC in contributing to overcoming situations in which motivational challenges require enhanced encoding of task-relevant variables.

Taken together, our observed result pattern supports theories suggesting the ACC contributes to information sampling as well as to controlling motivational effort during adaptive behavior. The results suggest that these functions of the ACC are recruited when the task requires overcoming motivational challenges from anticipating losses and when learning takes place in a cognitively demanding situation. We discuss the specific rationale for this conclusion in the following.

## ACC modulates the efficiency of information sampling

We found that ACC-TUS increased the duration of fixating objects prior to choosing an object in the loss contexts. Longer fixation durations are typically considered to reflect longer sampling of information from the foveated object. This has been inferred from the longer fixational sampling of objects associated with higher outcome uncertainty [49] or that explicitly carry information that reduces uncertainty about outcomes [50,51]. We, therefore, consider fixational sampling of visual objects to index information sampling. Consistent with this view, we found that ACC-TUS prolonged information sampling at high cognitive load when objects varied in more features from trial to trial and hence carried higher uncertainty about which feature was linked to reward (**Figs 2D and 2E and** S7). These results suggest that TUS might disrupt or modulate the activity of neuronal circuitries in ACC that would have responded to the demand for information about the feature values by controlling the duration of gaze fixations on available objects. Such activity has been documented to be encoded in ACC [16–19]. These studies found that neurons in the ACC encoded the pre-learned value of objects when they were fixated [18] or peripherally attended [16,19]. Moreover, subpopulations of ACC neurons also encode the value of objects that are not yet fixated but are possible future targets for fixations [17]. Disrupting these signals with TUS in our study is a plausible reason causing uncertainty about the value of objects and for prolonging information sampling durations.

In our study, ACC_TUS prolonged information sampling most prominently in the gain-loss context when subjects were uncertain not only about the possible gains following correct choices, but additionally about the possible losses following incorrect choices (**Fig 3D and 3E**). This is consistent with recent findings of neurons in the intact ACC that fire stronger when subjects fixate on cues that reduce uncertainty about anticipated losses [27]. In that study, trial-by-trial variations of ACC activity during the fixation of punishment-predicting cues correlated with trial-by-trial variations of seeking information about how aversive a pending outcome will be [27]. In our study, disrupting this type of activity with transcranial ultrasound may thus have disrupted the efficacy of acquiring information about the loss association of features while fixating objects. According to this interpretation, the TUS-induced prolongation of fixation durations indicates that an intact ACC critically enhances the efficiency of visual information sampling to reduce uncertainty about aversive outcomes.

## ACC mediates feature-specific credit assignment for aversive outcomes

The altered information sampling after ACC-TUS was only evident in the gain-loss context and depended on experiencing losing tokens. We found that losing 1, 2, or 3 tokens significantly impaired performance in ACC-TUS compared to sham or striatum sonication (**Fig 3F**). This valence-specific finding might be linked to the relevance of the ACC for processing negative events and adjusting behavior after negative experiences. Imaging studies have consistently shown ACC activation in response to negatively valenced stimuli or events [33–35]. Behaviorally, the experience of loss outcomes is known to trigger autonomous vigilance responses that reorient attention and increase explorative behaviors [37,52,53]. Such loss-induced exploration can help avoid threatening stimuli, but it comes at the cost of reducing

the depth of processing the loss-inducing stimulus [36]. Studies in humans have shown that stimuli associated with loss of monetary rewards or aversive outcomes (aversive images, odors, or electrical shocks) are less precisely memorized than stimulus features associated with positive outcomes [54–56]. Such a reduced influence of loss-associated stimuli on future behavior may partly be due to loss experiences curtailing the engagement with those stimuli and reducing the evaluation of their role for future choices, as demonstrated, e.g., in a sequential decision-making task [57]. One behavioral consequence of a reduced evaluation or memorization of aversive or loss-associated stimuli is an over-generalization of aversive outcomes to stimuli that only share some resemblance with the specific stimulus that caused the loss experience. Such over-generalization is evolutionary meaningful because it can support the faster recognition of similar stimuli as potentially threatening even if these stimuli are not a precise instance of a previously encountered, loss-inducing stimulus [36,58]. For our task, such a precise recognition of features of a chosen object was pivotal to learning from feature-specific outcomes. Our finding of reduced learning from loss outcomes, therefore, indicates that ACC-TUS exacerbated the difficulty to assign negative outcomes to features of the chosen object.

Neurophysiological support for this interpretation of the ACC to mediate negative credit assignment comes from studies showing a substantial proportion of neurons in ACC encode feature-specific prediction errors [15]. These neurons responded to an error outcome most strongly when it was unexpected and when the chosen object contained a specific visual feature. This is precisely the information that is needed to learn which visual features should be avoided in future trials. Consistent with this importance for learning, the study also reported that feature-specific prediction error signals in the ACC predict when neurons updated value expectations about specific features [15,59]. Thus, neurons in ACC signal feature-specific negative prediction errors and the updating of feature-specific value predictions. It seems likely that disrupting these signals with TUS will lead to impaired feature-specific credit assignment in our task. This scenario is supported by the finding that ACC-TUS impaired flexible learning most prominently at high load, i.e., when there was a high degree of uncertainty about which stimulus feature was associated with gains and which features were associated with losses (**Fig 2C**). This finding suggests that outcome processes such as credit assignment in an intact ACC critically contribute to the learning about aversive distractors.

## A role of the ACC to determine learning rates for aversive outcomes

So far, we have discussed that ACC-TUS has likely reduced the efficiency of information sampling and this reduction could originate from the disruption of feature-specific credit assignment processes. These phenomena were exclusively observed in the loss context, and ACC-TUS did not change learning or information sampling in the gain-only context. Based on this finding, we propose that the ACC plays an important role in determining the learning rate for negative or aversive outcomes and thereby controls how fast subjects learn which object features in their environments have aversive consequences. Such learning from negative outcomes was particularly important for our task at higher cognitive load [60]. At high load, a single loss outcome provided unequivocal information that all features of the chosen object were non-rewarded features in the ongoing learning block. This was different for positive outcomes. Receiving a positive outcome was linked to only 1 feature out of 2 or 3 features of the chosen object, making it more difficult to discern the precise cause of the outcome. The higher informativeness of negative than positive outcomes can explain why ACC-TUS caused a selective learning impairment in the loss context when assuming that TUS introduced uncertainty in the neural information about the cause of the outcome. This interpretation is consistent with the established role of the ACC to encode different types of errors [32,61,62] and with

computational evidence that learning from errors and other negative outcomes is dependent on a dedicated learning mechanism separately of learning from positive outcomes [60,63–65].

The suggested role of the ACC to determine the rate of learning from negative outcomes describes a mechanism for establishing which visual objects are distractors and should be actively avoided and suppressed in a given learning context [66]. When subjects experience aversive outcomes, the ACC may use these experiences to bias attention and information sampling away from the negatively associated stimuli. This suggestion is consistent with the fast neural responses in ACC to attention cues that trigger inhibition of distracting and enhancement of target information [16,19]. In these studies, the fast cue-onset activity reflects a push-pull attention effect that occurred during covert attentional orienting and was independent of actual motor actions. This observation supports the general notion that ACC circuits are critical for guiding covert and overt information sampling during adaptive behaviors [18,25].

## Versatility of the focused-ultrasound protocol for transcranial neuromodulation

Our conclusions were made possible using a TUS protocol that was developed to interfere with local neural activity in deep neural structures which likely imposes a temporary functional disconnection of the sonicated area from its network [38,40]. We implemented an enhanced protocol that entailed quantifying the anatomical targeting precision (**S3A and S3B Fig**) and confirmed by computer simulations that sonication power reached the targets (**S3C–S3E Fig**). We also showed that the TUS pressure in ACC was comparative to the maximum pressure in the brain and noticeably higher than in other nearby brain structures such as the orbitofrontal cortex (see **Materials and methods** and **S3F Fig**). We also documented that the main behavioral effect (impaired learning) of ACC-TUS was evident relative to a within-task pre-sonication baseline and throughout the experimental behavioral session (**S9 Fig**) and that the main TUS-ACC effects were qualitatively consistent across monkeys (for monkey specific results, see **S1**–**S10 Figs**; there were no significant random effects of the factor *monkey* in the LME models). Importantly, the main behavioral effects of ACC-TUS in our task are consistent with the effects from more widespread, invasive lesions of the ACC in nonhuman primates. Widespread ACC lesions reduce affective responses to harmful stimuli [67], increase response perseverations [68], cause failures to use reward history to guide choices [7,8], and reduce control to inhibit prevalent motor programs [69,70]. Our study, therefore, illustrates the versatility of the TUS approach to modulate deeper structures such as the ACC that so far have been out of reach for noninvasive neuromodulation techniques such as transcranial magnetic stimulation (TMS) or transcranial direct current stimulation (tDCS) [71]. Despite these suggestions, it should be made explicit that there is so far a scarcity of insights about the specific effects of the TUS protocol on neural circuits. Therefore, future studies will need to investigate the neural mechanisms underlying the immediate and longer-lasting TUS effects on ACC neural circuits.

In summary, our results suggest that the ACC multiplexes motivational effort control and attentional control functions by tracking the costs of incorrect performance, optimizing feature-specific credit assignment for aversive outcomes, and actively guiding information sampling to visual objects during adaptive behaviors.

## Materials and methods

### Ethics statement

All procedures were in accordance with the National Institutes of Health Guide for the Care and Use of Laboratory Animals, the Society for Neuroscience Guidelines and Policies, and

approved by the Vanderbilt University Institutional Animal Care and Use Committee (M1700198-01).

## Experimental procedures

Two male macaque monkeys (monkey I 13.6 kg and monkey W 14.6 kg, 8 to 9 years of age) contributed to the experiments. They sat in a sound-proof booth in primate chairs with their head position fixed, facing a 21" LCD screen at a distance of 63 cm from their eyes to the screen center. Behavior, visual display, stimulus timing, and reward delivery were controlled by the *Unified Suite for Experiments* (USE), which integrates an IO-controller board with a unity3D video-engine-based control for displaying visual stimuli, controlling behavioral responses, and triggering reward delivery [72]. Prior to the ultrasound experiment, the animals were trained on the feature learning task in a kiosk training station [73]. Monkeys first learned to choose objects to receive an immediate fluid reward before a token system that provided animals with green circles per correct choice that symbolized tokens later cashed out for fluid reward. Tokens were presented above the chosen object and traveled to a token bar where they accumulated with successive correct performance. When 5 tokens were collected, the token bar blinked red/white, fluid was delivered through a sipper tube, and the token bar reset to 5 empty token placeholders (**Fig 1A**). The animals effortlessly adopted the token reward system as documented in [36]. Here, we used in separate blocks of 35 to 50 trials a condition with "gains-only" (3 tokens for correct choices, no penalties) and with "gains-and-losses" (2 tokens for correct choices and 1 token lost, i.e., removed from the token bar, for incorrect choices). The introduction of gains-and-losses effectively changed behavior. Animals learned slower, showed reduced plateau accuracy, enhanced exploratory sampling, and more checking of the token bar (**S2B–S2H Fig**).

## Task paradigm

The task required monkeys to learn feature-reward rules in blocks of 35 to 60 trials through trial-and-error by choosing 1 of 3 objects. Objects were composed of multiple features, but only 1 feature was associated with reward (**Fig 1A–1C**). A trial started with the appearance of a black circle with a diameter of 1 cm (0.5° radius wide) on a uniform gray background on the screen. Monkeys fixated the black circle for 150 ms to start a trial. Within 500 ms after the central gaze fixation registration, 3 objects appeared on the screen randomly at 3 out of 4 possible locations with an equal distance from the screen center (10.5 cm, 5° eccentricity). Each stimulus had a diameter of 3 cm (approximately 1.5° radius wide). To choose an object, monkeys had to maintain fixation onto the object for at least 700 ms. Monkeys then had 5 s to choose 1 of 3 objects or the trial was aborted. Choosing the correct object was followed by a yellow halo around the stimulus as visual feedback (500 ms), an auditory tone, and either 2 or 3 tokens (green circles) added to the token bar (**Fig 1A**). Choosing an object without the rewarded target feature was followed by a blue halo around the selected objects, a low-pitched auditory feedback, and in the loss conditions, the presentation of a gray "loss" token that traveled to the token bar where one already attained token was removed. The timing of the feedback was identical for all types of feedback. In each session, monkeys were presented with up to 36 separate learning blocks, each with a unique feature-reward rule. Across all 48 experimental sessions, monkeys completed on average 23.6 (±4 SE) learning blocks per session (monkey W: 20.5 ± 4, monkey I: 26.5 ± 4). For each experimental session, a unique set of objects was defined by randomly selecting 3 dimensions and 3 feature values per dimension (e.g., 3 body shapes: oblong, pyramidal, and ellipsoid; 3 arm types: upward pointy, straight blunt, downward flared; 3 patterns: checkerboard, horizontal striped, vertical sinusoidal; Watson and

colleagues, 2019 [74] have documented the complete library of features). Of this feature set, 3 different task conditions were defined: One task condition contained objects that varied in only 1 feature while all other features were identical, i.e., the object body shapes were oblong, pyramidal, and ellipsoid, but all objects had blunt straight arms and uniform gray color. A second task condition defined objects that varied in 2 feature dimensions ("2D" condition), and a third task condition defined objects that varied in 3 feature dimensions ("3D" condition). Learning is systematically more demanding with increasing number of feature dimensions that could contain the rewarded feature (for computational analysis of the task, see Womelsdorf and colleagues, 2021). We refer to the variations of the object feature dimensionality as cognitive load because it corresponds to the size of the feature space subjects have to search to find the rewarded feature (**Figs 1E and** S1), and 1D, 2D, and 3D conditions varied randomly across blocks.

## Experimental design

Each session randomly varied across blocks 2 motivational conditions (gain/loss and gains-only) and 3 cognitive load conditions (1D, 2D, and 3D). Randomization ensured that all (6) combinations of conditions were present in the first 6 blocks prior to sonication and that all combinations of conditions were equally often shown in the 24 blocks after the first 6 blocks. After monkeys completed the first 6 learning blocks (on average 12 min after starting the experiment), the task was paused to apply TUS. After bilateral placement of the transducer and sonication (or sham sonication) of the brain areas, the task resumed for 18 to 24 more blocks (**Fig 1D**). Experimental sessions lasted 90 to 120 min.

## Transcranial ultrasound stimulation (TUS)

For transcranial stimulation, we used a single element transducer with a curvature of 63.2 mm and an active diameter of 64 mm (H115MR, Sonic Concepts, Bothell, Washington, United States of America). The transducer was attached to a cone with a custom-made trackable arm. Before each session, we filled the transducer cone with warm water and sealed the cone with a latex membrane. A conductive ultrasound gel was used for coupling between the transducer cone and the shaved monkey's head. A digital function generator (Keysight 33500B series, Santa Rosa, California, USA) was used to generate a periodic burst of 30-ms stimulation with a resonate frequency of 250 kHz and an interval of 100 ms for a total duration of 40 s and 1.2 MPa pressure (similar to [39,75]) (**Fig 1D**). A digital function generator was connected to a 150-watt amplifier with a gain of 55 dB in the continuous range to deliver the input voltage to the transducer (E&I, Rochester, New York, USA). We measured the transducer output in water using a calibrated ceramic needle hydrophone (HNC 0400, Onda Corp., Sunnyvale, California, USA) and created a linear relationship between the input voltage and peak pressure. To avoid hydrophone damage, only pressures below a mechanical index (MI) of 1.0 were measured and amplitudes above this were extrapolated. We have previously measured transducer output at MI > 1.0 with a calibrated optical hydrophone (Precision Acoustics, Dorchester, United Kingdom) to validate the linearity of this relationship at higher MI, but this calibrated device was not available during these studies. During stimulation, the bi-directionally coupled (ZABDC50-150HP+, Mini Circuits Brooklyn, New York, USA) feedforward and feedback voltage were monitored and logged using a Picoscope 5000 series (A-API; Pico Technology, Tyler, Texas, USA) and a custom written python script.

   Four different sonication conditions were pseudo-randomly assigned to the experimental days per week for a 12-week experimental protocol per monkey. These 4 conditions consisted of high energy TUS in anterior striatum (TUS-STR), high energy TUS in anterior cingulate

cortex (TUS-ACC), sham anterior striatum (Sham-STR), and sham anterior cingulate cortex (Sham-ACC) (**Fig 1D**). We sequentially targeted an area in both hemispheres (each hemisphere for a 40-s duration) with real-time monitoring of the distance of the transducer to the targeted area (**S3A Fig**) and monitoring of the feedforward power (**S3B Fig**). Sham conditions were identical to TUS conditions, only no power was transmitted to the transducer.

## Data analysis

**Trial-level statistical analysis.** We tested TUS effects on behavior at the trial level using LMEs models [10] with 4 main factors: *cognitive load (Cog$_{Load}$)* with 3 levels (1D, 2D, and 3D distractor feature dimensions, ratio scale with values 1, 2, and 3), *trial in block (TIB)*, *previous trial outcome* (*Prev$_{Outc}$*) which is the number of tokens gained or lost in the previous trial, motivational token condition, which we call the *motivational gain/loss context (MCtx$_{Gain/Loss}$)* with 2 levels (1, for the loss condition, and 2, for the gain condition, nominal variable), TUS condition (*TUS$_{Cnd}$*) with 4 levels (Sham-ACC, TUS-ACC, Sham-STR, and TUS-STR), and *time relative to stim (T2Stim)* with 2 levels (before versus after stimulation). We used 3 other factors as random effects, a factor *target features* (*Feat*) with 4 levels (color, pattern, arm, and shape), weekday of the experiment (*Day*) with 4 levels (Tuesday, Wednesday, Thursday, and Friday), and the factor *monkeys* with 2 levels (W and I). We used these factors to predict 3 metrics (*Metric*): accuracy (*Accuracy*), reaction time (*RT*), and information sampling (*Sample$_{Explr}$*). The LME is formalized as in Eq 1.

$$Metric = Cog_{Load} + TIB + Prev_{Outc} + MCtx_{Gain/Loss} + TUS_{Cnd} + T2Stim + (1|Day) + (1|Feat)$$
$$+ (1|Monkeys) + b + \varepsilon \tag{1}$$

**Block-level analysis of behavioral metrics.** We used LME models to analyze across blocks how learning speed (indexed as the "learning trial" (*LT*) at which criterion performance was reached), post-learning accuracy (*Accuracy*), and metrics for choice and information sampling were affected by TUS. In addition to the fixed and random effects factors used for the LME in Eq 1, we also included the factor block *switching condition* (*Switch$_{Cnd}$*) with 2 levels (intra- and extra-dimensional switch). The LME had the form of Eq 2:

$$Metric = Cog_{Load} + MCtx_{Gain/Loss} + TUS_{Cnd} + T2Stim + (1|Day) + (1|Monkeys) + (1|Feat)$$
$$+ b + \varepsilon. \tag{2}$$

We extended the model to test for interactions of *TUS$_{Cnd}$*, *MCtx$_{Gain/Loss}$*, and *T2Stim*:

$$Metric = Cog_{Load} + MCtx_{Gain/Loss} + MCtx_{Gain/Loss} \times TUS_{Cnd} \times T2Stim + (1|Day) + (1|Monkey)$$
$$+ (1|Feat) + b + \varepsilon, \tag{3}$$

for interactions of *T2Stim*, *Cog$_{Load}$*, and *TUS$_{Cnd}$*:

$$Metric = Cog_{Load} + MCtx_{Gain/Loss} + TUS_{Cnd} \times T2Stim \times Att_{Load} \times TUS_{Cnd} + (1|Day)$$
$$+ (1|Monkey) + (1|Feat) + b + \varepsilon, \tag{4}$$

and for interactions of *T2Stim*, *Cog$_{Load}$*, *MCtx$_{Gain/Loss}$*, and *TUS$_{Cnd}$*:

$$Metric = Cog_{Load} + MCtx_{Gain/Loss} + TUS_{Cnd} \times T2Stim \times Att_{Load} \times TUS_{Cnd} \times MCtx_{Gain/Loss}$$
$$+ (1|Day) + (1|Monkey) + (1|Feat) + b + \varepsilon. \tag{5}$$

## Supporting information

**S1 Text. Supplementary Information.** Contains detailed methods for precise TUS neuronavigation, TUS simulations, and data analysis.
(DOCX)

**S1 Fig. Cognitive load effect on learning and fixational information sampling.** (**A, B**) Both monkeys reached the learning criterion of 80% or more correct trials (based on a 12 trials forward-looking window). Learning is fastest at low cognitive load (*light gray*), and slowest at high cognitive load (*dark gray*). In all panels, the left column shows the results for monkey W, the middle for monkey I, and the right for both monkeys combined. (**C**) Post-learning accuracy is significantly reduced in higher cognitive load (LMEs, $P < 0.001$). (**D, E**) Information sampling is the duration in msec. of fixational sampling of objects prior to making a choice. Information sampling increased at the beginning of a block, reached a maximum during learning, and but remained elevated only at the highest cognitive load (LMEs, $p < 0.001$). (**F**) Choice reaction time is the time from stimulus onset to the onset of the final fixation (that chooses the object). It increased with cognitive load (Kruskal–Wallis test, $p < 0.001$). (**G**) Choice sampling is measured as the duration of sampling the chosen object prior to the final choice fixation. Choice sampling is increased in blocks with higher cognitive load (Kruskal–Wallis test, $p < 0.001$). (**H**) Asset sampling duration quantifies how long subjects fixate the token bar prior to choosing an object. Asset sampling is independent of cognitive load and more extensive in monkey I. Data show means and standard error of the mean. Data associated with this plot could be found at: https://figshare.com/projects/TUS_PlosBiology/144330.
(EPS)

**S2 Fig. Effects of gain-loss and gain-only learning contexts on learning and gaze sampling behaviors.** (A, B) Same format as **S1A and S1B Fig**. Learning is faster in the gain-only context (blue), than in the gain-loss context. In all panels, the left column shows the results for monkey W, the middle for the monkey I, and the right for both monkeys. (**C**) Post-learning accuracy is significantly reduced in the gain-loss context (Wilcoxon test, $P < 0.001$). (**D, E**) Information sampling reaches a higher maximum during learning in the gain-loss context (red) compared to the gain-only context (blue) (Wilcoxon test, $P < 0.001$). (**F**) Choice reaction time is slower in the gain-loss context (Wilcoxon test, $p < 0.001$). (**G**) Sampling of the object that is subsequently chosen is longer in the gain-loss context (Wilcoxon test, $p < 0.001$). (**H**) Asset sampling duration is longer in the gain-loss context (Wilcoxon test, $p < 0.001$). Data show mean and standard error of the mean. Data associated with this plot could be found at: https://figshare.com/projects/TUS_PlosBiology/144330.
(EPS)

**S3 Fig. Transcranial ultrasound stimulation localization, energy, and sonication focus specifications.** In each experimental session, we positioned the transducer sonication beam to focus on left and right hemisphere ACC and striatum and sonicated the area for 40 seconds in each hemisphere. (**A**) Both hemispheres in both monkeys were targeted precisely within an averaged sub-millimeter distance from the center of focal beam of the transducer to the anatomical target region. (**B**) Real-time monitoring of output voltage to the transducer confirmed reliable feedforward voltage range for both monkeys (see **Materials and methods**). (**C,D**) Computer simulations show a reliable range of RMS deviation values of negative peak pressure at the focus of the sonication attenuated at −3 dB (**C**), and −6 dB (**D**). (**E**) Maximum peak pressure at the targeted area. (**F**) For ACC-TUS sessions, to assure the effect is not attributed to OFC that appears to be aligned with the TUS orientation in **Fig 2A**, we compared the spatial average within a 4-mm cubic volume around each label target. Simulations showed ACC

received significantly higher compared with OFC (Wilcoxon test, $P < 0.0001$) and showed comparative value relative to the maximum pressure values in brain. Lines in panels (**A–D**) show standard error of the mean, and in (**F**) show median across all individual sessions (small dots). Data associated with this plot could be found at: https://figshare.com/projects/TUS_PlosBiology/144330. ACC, anterior cingulate cortex; OFC, orbitofrontal cortex; RMS, root mean squared; TUS, transcranial ultrasound stimulation.
(EPS)

**S4 Fig. Transcranial ultrasound stimulation effect on learning for individual monkeys (left and middle column) and their average (right column).** (**A**, **B**) Learning curves for 4 different experimental conditions: high energy TUS in ACC (TUS-ACC; red), or anterior striatum (TUS-STR; green), or sham ACC (Sham-ACC; darkened red), or sham anterior striatum (Sham-STR; darkened green). Learning curves are shallower after TUS-ACC in the gain-loss context (A) but not in the gain-only context (B). (C, D) The reduced learning speed (increased trials-to-criterion) with TUS-ACC in the gain-loss learning context (C) but not in the gain-only learning context (D). Detailed statistics are provided in S1 Table. Data show mean and standard error of the mean. The black asterisks show significant main effect of TUS conditions and a significant difference between the TUS and the baseline (pre-TUS) conditions (for the TUS condition underneath the asterisks). Horizontal black lines indicate significant pairwise difference between TUS conditions. Data associated with this plot could be found at: https://figshare.com/projects/TUS_PlosBiology/144330.
(EPS)

**S5 Fig. TUS interaction of cognitive load and motivational context.** (**A, B**) TUS in ACC reduces learning speed at higher cognitive load in the gain-loss learning context (**A**), but not the gain-only learning context (**B**) (for the full multiple comparison corrected statistical results see **S2 Table**). (**C**) Marginally normalized trials-to-criterion are significantly higher with TUS-ACC in the gain-loss learning context, and the effect in TUS-ACC was only significant at higher cognitive load conditions 2D and 3D (random permutation $p < 0.05$). Error bars are standard error of the mean. The white rectangle in (**C**) shows learning in a TUS condition is different from other TUS conditions. Each cell is color coded with the mean value ± standard error of the mean with a low to high value gradient from left to right. The white asterisk shows cognitive loads in a learning context in a TUS condition is significantly different from other TUS conditions. In all panels, the left column shows the results for monkey W, the middle for the monkey I, and the right for both monkeys combined. Black cross (×) and asterisk shows significant interaction of cognitive load and TUS conditions. Black asterisks show significant main effect of TUS conditions and a significant difference between the TUS and the baseline (pre-TUS) conditions (for the TUS condition underneath the asterisks). Data associated with this plot could be found at: https://figshare.com/projects/TUS_PlosBiology/144330.
(EPS)

**S6 Fig. Transcranial ultrasound stimulation effects on explorative behavior.** (**A, B**) *Information sampling* curves showing the average fixation durations on objects before making a choice. With the beginning of a new learning block *information sampling* increases to a maximum during learning and then slowly reduces to a baseline level. TUS in ACC in the gain-loss learning context (**A**), but not in the gain-only context causes elevated exploratory sampling (**B**). (**C, D**) *Information sampling* is increased with TUS in ACC relative to baseline and other TUS conditions in the gain-loss context (**C**) but not the gain-only context (**D**). (**E, F**) Post-learning accuracy is not significantly different from the baseline and other TUS conditions in the block-level analysis, neither in gain-loss context (**E**), nor the gain-only context (**F**) (detailed

statistics in **S1 Table**). The black asterisks show significant main effect of TUS conditions and a significant difference between the TUS and the baseline (pre-TUS) conditions (for the TUS condition underneath the asterisks). Horizontal black lines indicate significant pairwise difference between TUS conditions. Data associated with this plot could be found at: https://figshare.com/projects/TUS_PlosBiology/144330.
(EPS)

**S7 Fig. Distribution of gaze fixation duration before a choice.** (**A**) Distribution of gaze fixation duration on each object before making a choice (in brown) which we referred to as a measure of *information sampling*. In contrast to fixations indexing information sampling a "choice fixation" was registered when a fixation duration was ≥700 ms (in yellow). (**B–E**) Bootstrap distributions on information sampling for different cognitive load (**B**), motivational context (**C**), and TUS in gain-loss motivational context (**D**), and TUS in gain-only motivational context conditions (**E**). Data associated with this plot could be found at: https://figshare.com/projects/TUS_PlosBiology/144330.
(EPS)

**S8 Fig. Logistic regression GLM fit to performance accuracy.** (**A–D**) Average logistic regression fit using GLM on performance accuracy over learning blocks for different, (**A**), cognitive load, (**B**), motivational context, (**C**), TUS in gain-loss motivational context, and (**D**), TUS in gain-only motivational context conditions. Shaded error bars are standard errors of the mean. Data associated with this plot could be found at: https://figshare.com/projects/TUS_PlosBiology/144330. GLM, generalized linear mixed model; TUS, transcranial ultrasound stimulation.
(EPS)

**S9 Fig. Time course of the effect of transcranial ultrasound stimulation on learning and explorative behavior.** (**A**) Monkeys show a gradual deterioration of learning with TUS in ACC over time in the gain-loss conditions. (**B**) Information sampling increases over time relative to the time of TUS in ACC. The x-axis shows the time in minutes relative to TUS stimulation within a session. The time-step resolution is 20 min. Both monkeys show a similar time course with the slowing of learning from 20–40 min after the stimulation to the end of the session (randomized permutation test, $p < 0.05$). Shaded error bars are standard errors of the mean. Data associated with this plot could be found at: https://figshare.com/projects/TUS_PlosBiology/144330.
(EPS)

**S10 Fig. Session level effect of transcranial ultrasound stimulation on learning and information sampling.** Average results for each session and across sessions for monkey W and I, and both combined (left, middle, and right column, respectively). (**A, B**) TUS in ACC caused slower learning in both monkeys in the gain-loss context (**A**), but not in the gain-only context (**B**). (**C, D**) TUS in ACC causes longer information sampling than Sham STR in the gain-loss context (**C**). There was, however, no difference to baseline or in the gain-only context. (**D**). (**E, F**) TUS in ACC reduced post-learning accuracy relative to baseline and other TUS conditions in the gain-loss context (**E**), but not the gain-only context (**F**). For detailed statistics, see **S1 Table**. Data show session means for the pre-stimulation baseline blocks (*gray*) and across blocks after TUS/Sham (*colored*). Significant pairwise comparisons are indicated with black horizontal lines between each pair of conditions and the black asterisk shows significant session-level difference (post TUS versus baseline) of each behavioral measure. Data associated with this plot could be found at: https://figshare.com/projects/TUS_PlosBiology/144330.
(EPS)

**S11 Fig. Trial-level effects of transcranial ultrasound stimulation on post-outcome performance adjustment.** (**A**–**D**) Normalized performance accuracy in the 5 trials after receiving either a loss of 1 token (**A**), no loss after an incorrect response (**B**), a gain of 2 tokens (**C**), and a gain of 3 tokens (**D**). TUS in ACC (**red**) caused overall reduced performance accuracy in each monkey after losing 1 token or gaining 2 tokens that shows the effect is specific to the gain-loss context. (**E**) Both monkeys show reduced performance (y-axis) after TUS in ACC when the GTI was negative, i.e., then they on average had lost 1–3 tokens in the preceding 4 trials in gain-loss motivational context condition. (**F**) The performance accuracy was not different for any of GTI values in gain-only motivational context condition. Accuracy on the trial-level analysis is normalized by the mean and standard deviation of the accuracy of similar events in the baseline during the same TUS session. All statistics in these panels used randomization (permutation) tests with FDR correction of *p*-values for dependent samples with an alpha level of 0.05. The black asterisk shows points significantly different from pre-TUS baseline and relative to other TUS conditions. Data associated with this plot could be found at: https://figshare.com/projects/TUS_PlosBiology/144330. ACC, anterior cingulate cortex; GTI, gross token income; TUS, transcranial ultrasound stimulation.
(EPS)

**S1 Table.** Statistical results considering data from individual blocks (block level, left) and average data from individual sessions (session level, right) for trials-to-criterion (**A**, **B**), information sampling durations (**C**, **D**), and the post-learning plateau accuracy (**E**, **F**). Each table shows results of 3 different tests for the 2 motivational learning contexts: gain-only (G, in blue font) and gain-loss (GL, in red font). The *p*-values on the diagonal show Wilcoxon tests for each TUS condition compared to its baseline (before the stimulation). The non-diagonal cells show *p*-values for pairwise comparisons of each pair of the TUS conditions. The overall Kruskal–Wallis test results is shown on the right outside of each table. All *p*-values are FDR-corrected with an alpha level of 0.05, as explained in the methods. Data associated with this table could be found at: https://figshare.com/projects/TUS_PlosBiology/144330.
(EPS)

**S2 Table.** Statistical results split for low (left) and high (right) cognitive load conditions for trials-to-criterion (**A**, **B**), information sampling durations (**C**, **D**), and the post-learning plateau accuracy (**E**, **F**). Each table shows results of 3 different tests for the 2 motivational learning contexts: gain-only (G, in blue font) and gain-loss (GL, in red font). The *p*-values on the diagonal show Wilcoxon tests for each TUS condition compared to its baseline (before the stimulation). The non-diagonal cells show *p*-values for pairwise comparisons of each pair of the TUS conditions. The overall Kruskal–Wallis test results is shown on the right outside of each table. All *p*-values are FDR-corrected with an alpha level of 0.05, as explained in the methods. Data associated with this table could be found at: https://figshare.com/projects/TUS_PlosBiology/144330.
(EPS)

# Acknowledgments

The authors thank Dr. Marcus Watson for help with the experimental software, Huiwen Luo for assistance with calibrating optically tracked devices using MRI, and Adrienne Hawkes for technical assistance with voltage monitoring. The content is solely the responsibility of the authors and does not necessarily represent the official views of the National Institutes of Health.

## Author Contributions

**Conceptualization:** Kianoush Banaie Boroujeni, Charles F. Caskey, Thilo Womelsdorf.

**Data curation:** Kianoush Banaie Boroujeni.

**Formal analysis:** Kianoush Banaie Boroujeni.

**Funding acquisition:** Charles F. Caskey, Thilo Womelsdorf.

**Investigation:** Kianoush Banaie Boroujeni, Robert Louie Treuting, Charles F. Caskey, Thilo Womelsdorf.

**Methodology:** Kianoush Banaie Boroujeni, Michelle K. Sigona, Thilo Womelsdorf.

**Project administration:** Charles F. Caskey, Thilo Womelsdorf.

**Resources:** Charles F. Caskey, Thilo Womelsdorf.

**Software:** Kianoush Banaie Boroujeni, Michelle K. Sigona, Thomas J. Manuel, Charles F. Caskey, Thilo Womelsdorf.

**Supervision:** Charles F. Caskey, Thilo Womelsdorf.

**Validation:** Kianoush Banaie Boroujeni, Charles F. Caskey, Thilo Womelsdorf.

**Visualization:** Kianoush Banaie Boroujeni.

**Writing – original draft:** Kianoush Banaie Boroujeni, Charles F. Caskey, Thilo Womelsdorf.

**Writing – review & editing:** Kianoush Banaie Boroujeni, Charles F. Caskey, Thilo Womelsdorf.

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
