## [Editor Report · Decision Letter 0]

20 Jan 2022

Dear Dr Womelsdorf, 

Thank you for submitting your manuscript entitled "Transcranial Ultrasound Stimulation in Anterior Cingulate Cortex Impairs Information Sampling and Learning in Loss Contexts" for consideration as a Research Article by PLOS Biology.

Your manuscript has now been evaluated by the PLOS Biology editorial staff, as well as by an academic editor with relevant expertise, and I am writing to let you know that we would like to send your submission out for external peer review. Please accept my apologies for the delay in sending this decision to you.

Before we can send your manuscript to reviewers, we need you to complete your submission by providing the metadata that is required for full assessment. To this end, please login to Editorial Manager where you will find the paper in the 'Submissions Needing Revisions' folder on your homepage. Please click 'Revise Submission' from the Action Links and complete all additional questions in the submission questionnaire.

Once your full submission is complete, your paper will undergo a series of checks in preparation for peer review. Once your manuscript has passed the checks it will be sent out for review. To provide the metadata for your submission, please Login to Editorial Manager (https://www.editorialmanager.com/pbiology) within two working days, i.e. by Jan 24 2022 11:59PM.

If your manuscript has been previously reviewed at another journal, PLOS Biology is willing to work with those reviews in order to avoid re-starting the process. Submission of the previous reviews is entirely optional and our ability to use them effectively will depend on the willingness of the previous journal to confirm the content of the reports and share the reviewer identities. Please note that we reserve the right to invite additional reviewers if we consider that additional/independent reviewers are needed, although we aim to avoid this as far as possible. In our experience, working with previous reviews does save time. 

If you would like to send previous reviewer reports to us, please email me at ggasque@plos.org to let me know, including the name of the previous journal and the manuscript ID the study was given, as well as attaching a point-by-point response to reviewers that details how you have or plan to address the reviewers' concerns. 

Given the disruptions resulting from the ongoing COVID-19 pandemic, please expect some delays in the editorial process. We apologise in advance for any inconvenience caused and will do our best to minimize impact as far as possible.

Kind regards,

Gabriel

Gabriel Gasque

Senior Editor

PLOS Biology

ggasque@plos.org

---

## [Decision Letter · Decision Letter 1]

23 Mar 2022

Dear Dr Womelsdorf,

Thank you for submitting your manuscript "Transcranial Ultrasound Stimulation in Anterior Cingulate Cortex Impairs Information Sampling and Learning in Loss Contexts" for consideration as a Research Article at PLOS Biology. Please accept my sincere apologies for the long delays that you have experienced during the peer review process. I am now handling your submission since my colleague Gabriel is currently out of the office. Your manuscript has been evaluated by the PLOS Biology editors, an Academic Editor with relevant expertise, and by three independent reviewers.

The reviews are attached below. You will see that the reviewers are generally positive and find the manuscript interesting and well-done. However, they raise some overlapping concerns and ask that additional discussions about the limitations of the study should be included, as well as providing a clearer contextualization of the findings and the surrounding literature. In addition, Reviewer #3 raises concerns with the reporting of some of the statistical analyses and notes that an additional gain-loss only control experiment should be included.

In light of the reviews, we will not be able to accept the current version of the manuscript, but we would welcome re-submission of a much-revised version that takes into account the reviewers' comments. We cannot make any decision about publication until we have seen the revised manuscript and your response to the reviewers' comments. Your revised manuscript is also likely to be sent for further evaluation by the reviewers.

We expect to receive your revised manuscript within 3 months. Please email us (plosbiology@plos.org) if you have any questions or concerns, or would like to request an extension. At this stage, your manuscript remains formally under active consideration at our journal; please notify us by email if you do not intend to submit a revision so that we may end consideration of the manuscript at PLOS Biology.

**IMPORTANT - SUBMITTING YOUR REVISION**

*Re-submission Checklist*

*Published Peer Review*

*PLOS Data Policy*

*Blot and Gel Data Policy*

Sincerely,

Richard

Richard Hodge, PhD

Associate Editor, PLOS Biology

rhodge@plos.org

REVIEWS:

Reviewer #1: This study wants to answer long standing questions within the reward learning literature about the roles of striatum and anterior cingulate cortex. To do this they use an elegant combination of a novel neurostimulation, Transcranial ultrasound stimulation (TUS), with a learning paradigm and thoughtful behavioural analyses in macaque monkeys. 

Specifically, the authors target two structures known to be involved in reward learning and decision making, anterior cingulate cortex (ACC) and striatum (STR) with their interfering neurostimulation and compare them to each other as well as two sham conditions. 

Somewhat surprisingly, they find behavioural impairments only after ACC stimulation, not STR. Moreover, only when the learning requires the macaques to consider potential losses as well as gains, does an impairment emerge. 

When looking the data more carefully, they find that this is driven by only the data in which the animals need to ignore irrelevant dimensions, increasing the credit assignment problem. ACC stimulation, however does not only slow learning to criterion, but also reduces accuracy after criterion and leads to clustered errors. 

Overall, the results are solid and very interesting to the field. I do not have any major suggestions regarding data analysis. However, some of the framing, particularly in the introduction and parts of the discussion is confusing. The authors spend a lot of time discussing different theories of ACC during decision making that only speak indirectly to their own results. Instead, there is a large ACC literature on reward learning and representations of different value dimensions (such as effort, loss and gains) as well as credit assignment that probably makes a more relevant framing.

I have tried to make my concerns and potential solutions as concrete as possible below.

Major comments:

1) The authors very strongly emphasize two distinct ideas about ACC. One holds that ACC is involved in value guided decision, particularly when it comes to assessing alternatives, foraging, search and prospective planning. The other, claims ACC is exclusively concerned the signalling and resolution of conflict through control. However, while both theories have implications on what might happen during value guided learning, both are primarily concerned with decision making processes. It is not immediately clear how either theory would explain the learning deficits in a straightforward manner without some additions to the original theories. In fact, the current findings, if anything, are in direct contradiction with Conflict theories (in its many forms) because the primary function of ACC should have been linked to the load manipulation not the extension to the loss domain! Newer theories of conflict do include effort as a component but by no means suggest that losses are equivalent to effort and or that effort supersedes their postulated roles of ACC in general conflict resolution and signalling of difficult. In fact, the authors suggestion that their gain-loss condition is about effort seems rather post hoc to link their results to those theories, rather than being an established measure of effort. The other theory they talk about is at least neutral towards their findings and could easily be extended to consider ACC to not only have a role in decision making but also learning when situations are sufficiently ambiguous (high load) and there are mixed incentives, making it important to take into account undesirable hypotheticals. Additionally, theories emphasizing environmental value, are highly consistent with the observed clustering effects of low performance after ACC interferences, as ACC has been proposed to trigger behavioural change/switching/leaving of environments as a reaction to negative feedback or depleting patches.

Additionally, there are previous studies by Kennerley & Wallis on multiplexing (doi: 10.1038/nn.2961 ) and Hayden and Platt ( 10.1126/science.1168488 ) on hypothetical reward and ACC, that are nicely consistent with their findings (and the authors already cite). 

To cut a long story short, their results and study are better placed within the literature of reward learning, multiplexing of different value dimension and credit assignment, as their study clearly a reward learning one. Of course, mentioning other theories can be valuable but it gets confusing when the introduction is set up to put two theories against each other that are not directly about the behavioural measures presented in the results. 

2) Given ACC stimulation pretty much only leads to an impairment in the most challenging conditions (losses and gains combined with high cognitive load/multiple dimensions), one alternative explanation that should be mentioned is that it is simply a matter of impairment under the most challenging conditions, in which time to criterion is highest anyway. The authors, have two convincing pieces of evidence against this in the negative feedback specific effects and the higher errors even after criterion (although that could also be because it is just harder to remember with distractors and losses), but there should maybe be a brief mention of this more generic explanation of the results. 

3) Post error adjustment is a strange term for their effect after ACC stimulation. Isn't is more like a lack of adjustment? While non stimulated animals only make occasional or random errors, ACC stimulation leads to clusters of errors that a characterized by an inability to improve performance even after negative feedback. Maybe a better term would be "error clustering", "low performance periods" or "an inability to escape low average reward/negative feedback through behavioural change".

4) One slightly odd feature of the task is that both choices and "explorative sampling" are based on fixation. That makes it quite difficult to interpret the change of fixation duration as a change in explorative sampling as it could also be linked to changed decision making. It also gives another reason why criterion might take longer with added dimensions because they might inadvertently pick an option when they first look at it when there are many possible features to observe. In short, the authors should probably be careful with their strong interpretation of viewing duration given the specific features of the task and mention why that measures is a bit complicated.

Minor comments:

A) The authors mention "Studies in humans have shown that stimuli associated with loss of monetary rewards or aversive outcomes (aversive images, odors, or electrical shocks) are less precisely memorized than stimulus features associated with positive outcomes (32-34)." There is also an interesting literature on planning and aversive pruning that might be relevant here. One example paper : doi:10.1371/journal.pcbi.1002410

B) In Figure 2D the cells look multi-coloured, which is a bit confusing. Is there are way to change the colour scheme?

C) Fig 3a unclear what the stat test is on. Is the only effect that comes out a test between aSTR stimulation and ACC stimulation?

D) I might be wrong here but it read as if the authors only do LME's for the plateau results. Is that correct and if so, why?

Reviewer #2: This study looks at the effects of focused ultrasound in the anterior cingulate cortex and anterior striatum in awake, behaving monkeys. It has a number of strengths - 1. There are multiple targets and the behavioral effects depend on which target was sonicated, 2. Real and sham sonications were interleaved, 3. The dispersion of ultrasound pressure was modeled in 3D. 4. The behavioral task was rich enough to differentiate attention, motivation, and learning. All facets of the study were well-executed. 

There are 2 drawbacks - 1. The ultrasound pressure appears to spread into orbitofrontal cortex. This could contribute the some of the effects. 2. The measurement and interpretation of some of the behavioral outputs, particularly, "information sampling," or "explorative sampling," is not well-described. Both of these issues can be addressed by simply clarifying the text, without further experiments or analysis.

Minor comments

It's helpful to include line numbers in the draft.

The effects of sonication are described variously as "stimulation," or "disruption," and are compared to a lesion. In other studies, ultrasound seems to enhance performance. Could the authors discuss why they think ultrasound is disruptive? Is this simply due to the current results, or are there other studies showing disruption?

The authors do not define "information sampling" nor do they specify the behavior that they interpret as information sampling. It seems like it has something to do with fixation duration. Is it just fixation duration or is there more to it?

It seems like learning rate was estimated by "trials to criterion" and, sometimes, linear extrapolation. A better approach is to fit a logistic regression with rate and asymptote parameters.

The idea that ultrasound effects are seen mainly in low-motivation conditions is consistent with results recently published by Munoz et al. in Brain Stimulation.

It isn't clear how many sonications and shams were done in each target region per monkey. Based on the supplement, it appears to be 6 sonications and 6 shams per target per monkey. Please confirm if this is correct.

Please report the absolute de-rated peak negative pressure.

Reviewer #3: The anterior cingulate cortex and the striatum are both implicated in learning and sequential decision-making, but causal evidence is sparse, especially in the primate. The study thus fills an important gap in understanding the causal links between activity in these structures and the process of learning feature-reward rules. The study applied transcranial ultrasound stimulation to these two structures and evaluated behavioral indices of learning in a task which manipulates motivational context (gain-only, gain-loss) and attentional and/or cognitive load (low, medium, high). After stimulation to anterior cingulate cortex in the gain-loss context, learning was slower, asymptotic accuracy was lower, and gaze duration before the choice was longer. The context-specificity of these effects are likely due to the fact that ACC stimulation selectively disrupted performance following losses, consistent with a large literature linking this structure to post-error compensatory processes. The fact that slower learning was driven by the higher cognitive load condition also resonates with older literature linking this region to adaptation in the face of conflict and cognitive demands. By providing causal evidence for long standing theories that link ACC to post-error and performance monitoring functions, this study could potentially be a valuable contribution to the literature. However, it would benefit from discussing this literature, including an additional control analysis, improving the precision of the text, a thorough copy edit, and an acknowledgement of a few of the limitations of the results and methods.

Major comments:

1) It is difficult to understand the results obtained with the use of linear mixed effect (LME) models. A few examples:

"Learning is slowed down (trials-to-criterion increased) after TUS in ACC in the gain-loss context (left, LME's, p=0.007) but not in the gain-only contexts (right, n.s.)." (Figure 2B)

"TUS in ACC but not sham-TUS in ACC or TUS or sham-TUS in the striatum (Figure 2A) changed this behavioral pattern. ACC-TUS selectively slowed learning in the gain-loss contexts compared to the gain-only contexts (LME's, t=2.67, p=0.007)" (p. 5)

Is this the interaction between the motivational context and the TUS condition? Please specify what term is being reported here and for all linear mixed effects models. Please also add the reports in other places, where it seemed that a linear mixed effect model was being used, but only the post hoc comparisons were reported (i.e. p.6, lines 8-11).

2) In causal NHP papers, it is standard to report effects individually for each animal, particularly when there are only 2 animals used. It seems that the effects reported here may not have been individually significant (though this information was not provided, at least in the main text). This doesn't necessarily mean that the results are not interpretable, but it is a limitation that should be commented on in the discussion.

3) Simulation was the only evidence provided for the efficacy of ultrasound stimulation and there was no experimental verification that the procedure changed the neural activity within the targeted structures. This is a major concern for ultrasonic stimulation since we still know little about its impact on neuronal activity. In the absence of some kind of experimental verification that the effects of stimulation were directional and localized in ACC (vs affecting fibers of passage or connected regions, for example), the authors should eliminate all inferences about the specificity and directionality of the effects (i.e. "ACC is necessary", "disrupting ACC", etc.). These claims occurred throughout the manuscript. The limitations in these causal interpretations also need to be made clear in the discussion. 

4) The loss-specificity of the results is an important part of the contribution of this paper, but the method used to show that ACC-TUS effects were specific to the losses appears to combine data in an odd way--combining the gain data between both contexts (one in which there was an effect and one in which there was not), rather than comparing gain and loss within the same gain-loss context. If I'm understanding this correctly, it would be helpful to perform an analysis analogous to Fig 3F and S9E for the gain-loss context only.

5) The loss-specificity and interaction with cognitive demands of the ACC-TUS stimulation resonate with a body of literature that links ACC to performance/error monitoring and compensatory functions. However, there was no discussion of these classic ideas and theories. It would be very helpful to situate these results in a proper historical context, rather than focusing the literature review on recent papers with tenuous relevance to these results.

Minor issues:

- When discussing the contextual change introduced by gain-only and gain-loss conditions, the terms "reward structure", "motivational/affective demands" and "payoff" are used interchangeably. This made it harder than necessary to understand the actual manipulations in the task. The interchangeability of these various descriptors also meant that the results occasionally veered off into speculative directions where the logic was tough to follow.

- The use of the term "exploration" in this paper was confusing. In the context of sequential decision-making tasks like this one, exploration has a technical meaning: a kind of non-exploitative decision focused on learning about the options. However, manuscript sometimes uses "explorative sampling" to refer to gaze duration. (Please note that the manuscript also sometimes used the term "fixational sampling" directly to talk about "explorative sampling" [e.g. p. 9].) It would be more clear to replace "explorative sampling" with a less loaded term, like "information sampling", as is already done in some parts of the text. This is particularly important because the text does refer to the explore/exploit dilemma in the traditional way elsewhere (i.e. p. 6).

- The term "attentional load" seems misplaced as the role of attention in this task is not clear. Increasing the number of stimulus features could increase load at a variety of cognitive levels, not just at the level of attention, and none of the analyses presented here suggest that this manipulation alters attention. Maybe "cognitive load" would be better?

- In the text, TUS conditions are called ACC-TUS, ACC-sham, aSTR-TUS, aSTR-sham, but in the figures they are called H-ACC, S-ACC, H-aSTR, S-aSTR. This second set of abbreviations is only explained in the supplement (Sham-ACC, High-ACC, Sham-STR, High-STR). 

- Notation was inconsistent for a variety of other terms throughout the manuscript and supplement (e.g. "LMEs" and "LME's", "t" and "t-stat", "gain/loss condition" and "gain-loss condition", "STR" and "aSTR").

- Definitions for some acronyms were missing entirely from the main text and found only in the supplement.

- Mean values of two major dependent variables are missing in the text (plateau accuracy and information sampling), though mean values of learning time are reported.

- In the Supplement: "Where appropriate we used a non-parametric approach by fitting generalized linear mixed effects models (GLME's)" [...] "we used a link Identity function, and a Poisson distribution for the response variable." A model that makes an assumption about the distributions is not non-parametric. Also, a log link would be more common for a Poisson distribution because the distribution is undefined for negative parameters. The use of a non-standard link function should always be justified.

- p. 5 last paragraph "(LME's, t=2.67, p=0.007))" - extra parenthesis

- p. 6 first paragraph - reference to Figure 3C,D. Is this actually referencing Figure 2C,D?

- The use of the acronyms GLME and LME was a bit unusual. I couldn't find their definitions in the main text, but in the supplement, they were defined as "linear mixed effect models (LME)", "generalized linear mixed effects models (GLME's)". Typically these would be abbreviated as "Linear Mixed Effects (LME) models" or else could be extended to "LMEMs" or "GLMEMs" for brevity.

- Figure 1: the small table presenting motivational contexts and axes presenting task dimensions are not referenced by the caption and the panel is unlabeled.

- Figure 2B: it is not clear what is meant by the horizontal lines and the star.

- Figure 2C: what is represented by * and X ?

- Figure 2D: what does "marginally normalized trials-to-criterion" mean? (labeled "Diff. Norm. Trial-to-criterion" in the figure)

- Figure 3: it might help to maintain a common convention with the Figure 2 (i.e. gain-loss and gain-only results could be presented under the same letter). There are also more ambiguous symbols here that are not defined in the caption.

- Figure S4A - typo in the title ("gain-loss")

- Figure S5A - significance marked for monkey W but not in both monkeys together? Is this different from Figure 2C presenting the same result?

- Figure S6E - missing significance star in both monkeys? Is this different from Figure 3C presenting the same result?

---

## [Decision Letter · Decision Letter 2]

25 Jul 2022

Dear Dr Womelsdorf,

Thank you for your patience while we considered your revised manuscript "Transcranial Ultrasound Stimulation in Anterior Cingulate Cortex Impairs Information Sampling and Learning in Loss Contexts" for publication as a Research Article at PLOS Biology. This revised version of your manuscript has been evaluated by the PLOS Biology editors, the Academic Editor and the original reviewers.

Based on the reviews, we are likely to accept this manuscript for publication, provided you satisfactorily address the remaining editorial requests (just below) and journal data and policy-related requirements detailed at the bottom of this email. Please note that all of these points must be addressed before we can move forward with your study.

TITLE: We'd would like you to consider an alternative title for this work that we feel will make the study more broadly accessible to our readership:

The anterior cingulate cortex is involved in flexible learning under motivationally challenging and cognitively demanding conditions

We appreciate that this removes information on the type of stimulation used, but thought that it allows the interesting biology to be more apparent.

ABSTRACT EDITS needed:

1) As per journal policy, the abstract needs to indicate that this work has been done in non-human primates. 

2) We also ask that you do some light textual editing of the abstract to correct the English language errors in the key sentence that sets up the question they are asking. 

The key sentence is missing "subjects":

“Information about feature values in ACC or STR might contribute to adaptive learning by guiding choices and information sampling to relevant objects, or they might have a more indirect, motivational function by allowing SUBJECTS to estimate the value of putting effort choosing objects.”

While you are making this correction, we'd also suggest you pass the abstract by a few colleagues not in your field. As PLOS Biology is a broad biology journal, your study will get more attention and interest if the abstract is clearly accessible to both neuroscientists and non-neuroscientists. 

METHODS:

Please move the key methods sections necessary for readers to evaluate your work into the main paper, rather than having them all in supplemental materials. This will allow readers to more easily assess your study.

We expect to receive your revised manuscript within two weeks. 

*Published Peer Review History*

*Press*

Sincerely,

Kris

Kris Dickson, Ph.D. (she/her)

Neurosciences Senior Editor/Section Manager,

kdickson@plos.org,

PLOS Biology

ETHICS STATEMENT: Requires additional detail.

Currently says:

“All procedures were in accordance with the National Institutes of Health Guide for the Care and Use of Laboratory Animals, the Society for Neuroscience Guidelines and Policies, and approved by the Vanderbilt University Institutional Animal Care and Use Committee (M1700198-01).”

As per our NHP policy, “Non-human primate studies must be performed in accordance with the recommendations of the Weatherall report “The use of non-human primates in research”. Manuscripts describing research involving non-human primates must include details of animal welfare, including information about housing, feeding, and environmental enrichment, and steps taken to minimize suffering, including use of anesthesia and method of sacrifice if appropriate. 

Our policy can be found at: https://journals.plos.org/plosbiology/s/animal-research#loc-non-human-primates”

Please ensure that these additional details are provided.

DATA POLICY:

ALL DATA must either be contained in the paper or openly available via a static public database. We cannot accept sole deposition of data or code to GitHub or a similar non-static site (https://journals.plos.org/plosbiology/s/data-availability). We require deposition to a static site, like Zenodo, FigShare, OSF. As your existing data is on a publicly available GitHub site, once that data is updated per our formatting requirements (see below), it can be copied to Zenodo. See the process for doing this here: https://docs.github.com/en/repositories/archiving-a-github-repository/referencing-and-citing-content. Once you do this, it will also generate a DOI number that you can provide us with.

Note that we do not require all raw data. Rather, we ask that all individual quantitative observations that underlie the data summarized in the figures and results of your paper be made available. Please ensure that you provide the individual numerical values that underlie the summary data displayed in the following figure panels as they are essential for readers to assess your analysis and to reproduce it:

Fig1FG; Fig2BCD; Fig3A-F

Supplemental Fig1A-H; Fig2A-H; Fig3A-F; Fig4A-D; Fig5A-C; Fig6A-F; Fig7A-E; Fig8A-D; Fig9AB; Fig10A-F; Fig11A-F

NOTE 2: Please also ensure that FIGURE LEGENDS in your manuscript include information on where the underlying data can be found, and ensure your supplemental data file/s has a legend. (NG: This step is often forgotten!!!)

Code Availability – 

We ask that you provide open access to the custom MATLAB code used in this work. As per our open access policies, archives should provide a public repository of the described software. The repository must have been in existence for over five years or be hosting more than 1,000 projects. Our policies on Code sharing are here: https://journals.plos.org/plosbiology/s/materials-software-and-code-sharing.

SPECIES INDICATED IN THE ABSTRACT? Required (and already mentioned above)

- Please note that per journal policy, the model system/species studied should be clearly stated in the abstract of your manuscript. 

DATA NOT SHOWN?

- Please note that per journal policy, we do not allow the mention of "data not shown", "personal communication", "manuscript in preparation" or other references to data that is not publicly available or contained within this manuscript. Please check your submission for any such statements and either remove mention of such data or provide figures presenting the results and the data underlying the figure(s).

Reviewer remarks:

Reviewer's Responses to Questions

PLOS authors have the option to publish the peer review history of their article (what does this mean?). If published, this will include your full peer review and any attached files.

Reviewer #1: No

Reviewer #2: No

Reviewer #3: No

Reviewer #1: The authors have addressed all my comments. My congratulations on a great study!

Reviewer #2: Thank you for responding to my concerns. I have no further comments.

Reviewer #3: The authors have largely addressed my concerns and I am comfortable endorsing publication at this time.

Sorry for the delay in returning this review. I hope you are all enjoying a nice summer.

---

## [Editor Report · Decision Letter 3]

9 Aug 2022

Dear Dr Womelsdorf,

Thank you for the submission of your revised Research Article "The anterior cingulate cortex causally supports flexible learning under motivationally challenging and cognitively demanding conditions" for publication in PLOS Biology. On behalf of my colleagues and the Academic Editor, Ben Seymour, I am pleased to say that we can in principle accept your manuscript for publication, provided you address a 2 editorial issues any remaining formatting and reporting issues. The formatting and reporting issues will be detailed in an email you should receive within 2-3 business days from our colleagues in the journal operations team; no action is required from you until then. Please note that we will not be able to formally accept your manuscript and schedule it for publication until you have completed any requested changes.

When making any corrections requested by our operations team, please also see to the following:

1) Please correct your meta-data files. We appreciate the provision of your Excel datasheet and the behavioral data in FigShare, but note that the data is currently not provided in such a way as to allow readers to see the underlying summary data that went into the creation of each graph in your submission. Please update the files such that summary data used to create these data panels in the main and supplemental figures are listed in separate tabs that are clearly labeled for each figure and supplemental figure (i.e. Fig1F, Fig1G...SuppFig1A...) . 

Note that we do not require all raw data. Rather, we ask that all individual quantitative observations that underlie the data summarized in the figures and results of your paper be made available. The numerical data provided should include all replicates AND the way in which the plotted mean and errors were derived (it should not present only the mean/average values).

2) Minor point - please replace "to" with "into" in the abstract:

"of putting effort INTO choosing objects."

Finally, please take a minute to log into Editorial Manager at http://www.editorialmanager.com/pbiology/, click the "Update My Information" link at the top of the page, and update your user information to ensure an efficient production process.

PRESS

Sincerely, 

Kris

Kris Dickson, Ph.D. (she/her)

Neurosciences Senior Editor/Section Manager

PLOS Biology

kdickson@plos.org